# Introducing Routing Uncertainty
# in Capsule Networks

**Fabio De Sousa Ribeiro**$^\nabla$      **Georgios Leontidis**$^\dagger$      **Stefanos Kollias**$^\nabla$

$^\nabla$ **Machine Learning Group**      $^\dagger$ **Department of Computing Science**
University of Lincoln, UK      University of Aberdeen, UK
`{fdesousaribeiro,skollias}@lincoln.ac.uk`      `georgios.leontidis@abdn.ac.uk`

## Abstract

Rather than performing inefficient *local* iterative routing between adjacent capsule layers, we propose an alternative *global* view based on representing the inherent uncertainty in part-object assignment. In our formulation, the *local* routing iterations are replaced with variational inference of part-object connections in a probabilistic capsule network, leading to a significant speedup without sacrificing performance. In this way, *global* context is also considered when routing capsules by introducing *global* latent variables that have direct influence on the objective function, and are updated discriminatively in accordance with the minimum description length (MDL) principle. We focus on enhancing capsule network properties, and perform a thorough evaluation on pose-aware tasks, observing improvements in performance over previous approaches whilst being more computationally efficient.

## 1 Introduction

Although capsule networks (CapsNets) have taken on a few different forms since their inception [1, 2, 3, 4], they are generally built upon the following core assumptions and premises:

(i) Capturing *equivariance* w.r.t. viewpoints in neural activities, and *invariance* in the weights;

(ii) High-dimensional coincidences are effective feature detectors;

(iii) Viewpoint changes have *nonlinear* effects on pixels, but *linear* effects on object relationships;

(iv) Object parts belong to a single object, and each location contains at most a single object.

In theory, a perfect instantiation of the above premises could yield more sample efficient models, that leverage robust representations to better generalise to unseen cases. Unlike current methods, humans can extrapolate object appearance to novel viewpoints after a single observation. Evidence suggests that this is because we impose coordinate frames on objects [5, 6]. Capsules imitate this concept by representing neural activities as poses of objects w.r.t. a coordinate frame imposed by an observer, and attempt to disentangle salient features of objects into their composing parts. This is reminiscent of inverse graphics [7], but is not explicitly enforced in capsule formulations since the learned pose matrices are not constrained to interpretable geometric forms. Another argument for CapsNets, is one that views capsules as an extension to the very successful inductive biases already present in CNNs, by wiring in some additional complexity to deal with viewpoint changes. One of the desired effects is to align the learned representations with those perceptually consistent with humans, which would also make adversarial examples less effective [8]. The additional complexity comes from replacing scalar neurons with vector valued neural activities, along with a high-dimensional coincidence filtering algorithm to detect capsule level features, known as capsule *routing* [2, 3]. This procedure is typically iterative, local and inefficient which has prompted further research on the topic [9, 10, 11, 12, 13].

### 1.1 Motivation & Contribution

**Weaknesses of Capsule Networks.** The memory bottleneck incurred by vector valued activations in addition to the iterative nature of capsule routing algorithms results in inefficient models. They are also prone to underfitting or overfitting if the number of routing iterations isn't properly set [2, 3]. To address the above weaknesses one may decide to naively replace the iterative nature of capsule routing with some faster alternative. However, to stay true to the premises of CapsNets, we argue that the four following points are of paramount importance for the research community to consider, when proposing algorithmic variants of CapsNets or capsule routing going forward:

(i) Whether **viewpoint-invariance** and **affine** transformation robustness properties are retained;

(ii) Changes in assumptions about part-object relationships are made explicit;

(iii) Whether capsules are still activated based on high-dimensional coincidences;

(iv) How do we handle the intrinsic **uncertainty** in assembling parts into objects.

Changes in the core assumptions of CapsNets aren't always made clear in recent literature, but emerge incidentally via the proposed modifications. This leads to ambiguities regarding what qualifies as a capsule network, which can make comparisons between methods more difficult and hinder progress. In this paper, we focus on the core premises of capsule networks, and on enhancing their advantages over CNNs: viewpoint-invariance, and affine transformation robustness whilst being more efficient.

**Contribution.** Rather than performing *local* iterative routing between adjacent capsule layers which is inefficient, we propose an alternative *global* view based on representing the inherent uncertainty in part-object relationships, by approximating a posterior distribution over part-object connections. Sources of uncertainty in assembling objects via a composition of parts can arise from numerous sources, such as: (i) feature occlusions due to observed viewpoints; (ii) sensory noise in captured data; (iii) object symmetries for which poses may be ambiguous such as spherical objects/parts.

In our formulation, the *local* routing iterations are replaced with variational inference of part-object connections in a probabilistic capsule network, leading to a significant speedup (Figure 4). In this way, we encourage *global* context to be taken into account when routing information, by introducing *global* latent variables which have direct influence on the objective function, and are updated discriminatively in accordance with the minimum description length (MDL) principle [14, 15]. Our experiments demonstrate that *local* iterative routing can be replaced by variational posterior inference of part-object connections in a *global* context setting, allowing the model to leverage the inherent uncertainty in assembling objects as a composition of parts to improve performance on pose-aware tasks.

## 2 Background: Capsule Networks

**Capsules.** A capsule $\mathbf{c}$ is a set of neurons $\mathbf{c} = \{a, \mathbf{M}\}$. Each capsule is composed of either a vector $\mathbf{m} \in \mathbb{R}^d$ or matrix $\mathbf{M} \in \mathbb{R}^{\sqrt{d} \times \sqrt{d}}$ of neurons, and an activation probability $a$. A single capsule is wired to represent a single entity, and its vector/matrix may learn to encode its pose w.r.t the coordinate frame imposed by an observer. The activation $a$ simply represents an entity's presence. A capsule network is composed of two or more capsule layers, with multiple capsules $N$ in each layer. Capsule routing takes place between adjacent capsule layers, i.e. $N_i$ capsules in a lower layer $\ell_i$ are routed to $N_j$ capsules in a higher layer $\ell_j$, which can be seen as a form of cluster finding. Contextually, capsules in $\ell_i$ are referred to as *parts* of objects (datapoints), and capsules in $\ell_j$ are *objects* (clusters). Each part capsule uses its relationship to the viewer (pose), to posit a vote for what the pose of the object it is part of should be. To achieve this, part capsule poses $\mathbf{M}_i$ are multiplied with trainable viewpoint-invariant, affine transformation weight matrices:

$$\mathbf{V}_{j|i} = \left\{ \mathbf{M}_i \cdot \mathbf{W}_{ij} \mid \forall \mathbf{c}_i \in \ell_i \,,\, \forall \mathbf{c}_j \in \ell_j \right\}, \quad \mathbf{W}_{ij} \in \mathbb{R}^{\sqrt{d} \times \sqrt{d}}. \tag{1}$$

where $\mathbf{V}_{j|i}$ denotes the $i^{\text{th}}$ part capsule vote for the $j^{\text{th}}$ object capsule pose, and $\mathbf{W}_{ij}$ are the weights,

**Inducing Nonlinearity.** Capsule poses $\mathbf{M}$ are not directly activated via nonlinear mappings but are compositions of affine/projective linear transformations, that increase in complexity as we traverse through the network. Nonlinearity is induced by the choice of routing algorithm [2, 3], and the vote agreement measure used in calculating the activation probability $a_j$ for each capsule $\mathbf{c}_j \in \ell_j$.

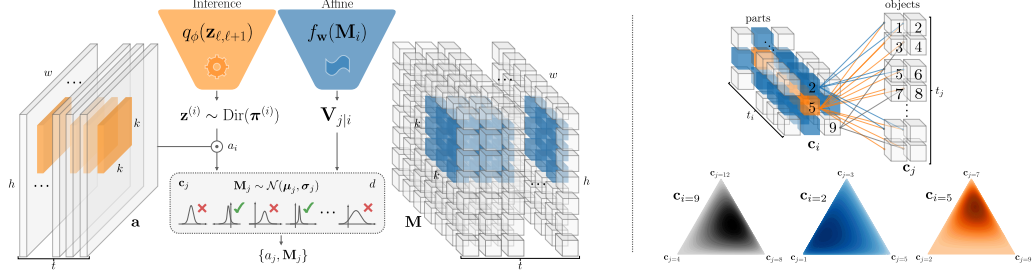

Figure 1: Our inference procedure in a given capsule layer (Left). Small example of part-object connections in convolutional voting for $k = 2$, drawn randomly from Dirichlet distributions (Right).

# 3 Uncertainty in Capsule Routing

Let $\mathcal{D}$ denote a set of data given as $m$ pairs $\{\mathbf{x}_i, y_i\}_{i=1}^m$, where $\mathbf{x}_i \in \mathbb{R}^d$ denotes a datapoint, and $y_i \in \{1, \ldots, K\}$ its corresponding label. Let $\mathbf{z}$ denote some latent variables associated with our observations $(\mathbf{x}, \mathbf{y})$, that capture underlying structure in our data $\mathcal{D}$ and help govern its distribution.

## 3.1 Defining Part-Object Connections

**Dense & Convolutional Voting.** In *dense* capsule voting, all part capsules are connected to all object capsules in the layer above. That is, each part capsule $\mathbf{c}_i \in \ell_i$ votes $N_j$ times, therefore each object capsule $\mathbf{c}_j \in \ell_j$ receives $N_i$ votes. The part-object connections are then $\mathbf{z}_{\ell_i, \ell_j} \in \mathbb{R}^{N_i \times N_j}$. Alternatively, in a convolutional capsule layer with kernel size $k$ and stride $s$, the number of object capsules that each part capsule $\mathbf{c}_i$ can vote for $N_{i \to j}$, is bounded above and below by

$$0 \leq N_{i \to j} \leq t_j \times \left\lceil \frac{k}{s} \right\rceil^2, \quad \text{and} \quad \mathbf{z}_{\ell_i, \ell_j}^{(i)} \in \mathbb{R}^{N_{i \to j}} \quad \forall \mathbf{c}_i \in \ell_i, \tag{2}$$

where $\lceil \cdot \rceil$ denotes the ceiling function, and $t_j$ denotes the number of output object capsule types, which are analogous to output channels in CNNs. Importantly, part capsules on the edge of feature maps vote for fewer objects than those in the middle (Figure 1), a fact which is very often overlooked in capsule research, leading to improper normalisation over objects and competition between capsules.

**Stochastic Variational Inference.** To represent our uncertainty about part-object relationships in a CapsNet, we look to approximate the (intractable) posterior distribution $p(\mathbf{z}|\mathcal{D})$ over part-object connections $\mathbf{z}$, with a chosen parameterised distribution $q_\phi(\mathbf{z}|\mathcal{D}) \approx p(\mathbf{z}|\mathcal{D})$ via variational inference (VI). In general, $q_\phi(\mathbf{z}|\mathcal{D})$ is optimised by updating the parameters $\phi$ such that the Kullback-Leibler (KL) divergence $D_{\mathrm{KL}}(q_\phi(\mathbf{z}|\mathcal{D}) \,||\, p(\mathbf{z}|\mathcal{D}))$ is minimised [15, 16, 17]. Next, we discuss and consider the inference of $q_\phi(\mathbf{z}|\mathcal{D})$ under two main modelling paradigms: *generative* and *discriminative*.

**Generative.** Under generative frameworks, a set of *local* latent variables $\mathbf{z}$ in models of the form $p_\theta(\mathbf{x}, \mathbf{z}) = p_\theta(\mathbf{x}|\mathbf{z})p(\mathbf{z})$ are often employed, such as in the variational autoencoder (VAE) [18]. Specifically, latent variables $\mathbf{z} = \{\mathbf{z}_i\}_{i=1}^m$ are inferred for each $\mathbf{x} = \{\mathbf{x}_i\}_{i=1}^m$, and maximum likelihood (ML) or maximum a posteriori (MAP) inference is performed on *global* parameters. The model is fit by maximising the Evidence Lower BOund (ELBO) on the marginal log-likelihood

$$\log p(\mathbf{x}) \geq \sum_{i=1}^m -D_{\mathrm{KL}}(q_\phi(\mathbf{z}_i|\mathbf{x}_i) \,||\, p(\mathbf{z}_i)) + \mathbb{E}_{q_\phi(\mathbf{z}_i|\mathbf{x}_i)}[\log p_\theta(\mathbf{x}_i|\mathbf{z}_i)] \triangleq \mathcal{L}_{\mathrm{local}}(\phi, \theta). \tag{3}$$

**Discriminative.** Under the discriminative framework, *global* latent variables $\mathbf{z}$ are often utilised and are shared among datapoints $\{\mathbf{x}_i\}_{i=1}^m$, for instance when inferring the posterior on the weights of a neural network (NN) [15, 17, 19]. The bound is on the conditional marginal log-likelihood

$$\log p(\mathbf{y}|\mathbf{x}) \geq \sum_{i=1}^m -\frac{1}{m} D_{\mathrm{KL}}(q_\phi(\mathbf{z}) \,||\, p(\mathbf{z})) + \mathbb{E}_{q_\phi(\mathbf{z})}[\log p(\mathbf{y}_i|\mathbf{x}_i, \mathbf{z})] \triangleq \mathcal{L}_{\mathrm{global}}(\phi) \tag{4}$$

To facilitate comparisons with the majority of research on CapsNets, we focus on the development and evaluation of our method in a discriminative setting. Formally, we are interested in estimating the conditional likelihood $p(\mathbf{y}|\mathbf{x}, \mathbf{z}) = \prod_{i=1}^m p(\mathbf{y}_i|\mathbf{x}_i, \mathbf{z})$ using probabilistic capsule network models.

## 3.2 Posterior Inference of Part-Object Connections

**Inference & Model Assumptions.** Using stochastic VI tools, we intend to find the best approximation $q_\phi^\star(\mathbf{z})$ that minimises $D_{\mathrm{KL}}(q_\phi(\mathbf{z}) \,||\, p(\mathbf{z}|\mathcal{D}, \mathbf{W}))$, where $\mathbf{z}$ are global latent part-object connection variables, and $\mathbf{W}$ are viewpoint-invariant transformation parameters, in a CapsNet with $L$ layers. We place a prior $p(\mathbf{z}^{(i)})$ over each part capsule's $\mathbf{c}_i \in \ell$ connections to the objects they vote for $\mathbf{c}_j \in \ell + 1$, and make the following factorised independence assumptions across capsule layers:

$$\mathbf{z}^{(i)} = (z_1, z_2, \ldots, z_{N_{i \to j}}) \sim p(\mathbf{z}^{(i)}) \quad \forall \mathbf{c}_i \in \ell_i, \qquad p(\mathbf{z}) = \prod_{\ell=1}^{L-1} \prod_{i=1}^{N_i} p(\mathbf{z}_\ell^{(i)}). \qquad (5)$$

We then make a variational approximation $q_\phi(\mathbf{z}_{\ell,\ell+1})$ to the posterior on part-object connection variables between adjacent capsule layers $\ell$ and $\ell + 1$, for all capsule layers in the network. Our model's likelihood $p(\mathcal{D}|\mathbf{z}, \mathbf{W})$, and mean-field variational family $q_\phi(\mathbf{z})$ are given by

$$p(\mathcal{D}|\mathbf{z}, \mathbf{W}) = \prod_{i=1}^{m} p(\mathbf{y}_i|\mathbf{x}_i, \mathbf{z}, \mathbf{W}), \qquad q_\phi(\mathbf{z}) = \prod_{\ell=1}^{L-1} \prod_{i=1}^{N_i} q_\phi(\mathbf{z}_{\ell,\ell+1}^{(i)}). \qquad (6)$$

The model is defined hierarchically where the object capsules in $\ell$ are the parts of $\ell + 1$, and so forth.

**Free Energy Objective.** The model is fit end-to-end by maximising the following lower bound on the conditional marginal log-likelihood $\log p(\mathbf{y}|\mathbf{x})$, which approximates its description length:

$$\mathcal{L}(\mathbf{y}|\mathbf{x}; \phi) \triangleq -\sum_{\ell=1}^{L-1} D_{\mathrm{KL}}(q_\phi(\mathbf{z}_{\ell,\ell+1}) \,||\, p(\mathbf{z}_\ell)) + \sum_{i=1}^{m} \mathbb{E}_{q_\phi(\mathbf{z})}[\log p(\mathbf{y}_i|\mathbf{x}_i, \mathbf{z}, \mathbf{W})]. \qquad (7)$$

In the general case, we perform variational inference on the part-object connection latent variables $\mathbf{z}$, and ML/MAP inference on $\mathbf{W}$. We find this to work well enough in practice, whilst significantly reducing the number of parameters needed and assumptions made, which is especially important in CapsNets given that efficiency is a major concern. Nonetheless, for full posterior learning, we can make one further mean-field assumption by: $q_{\phi,\theta}(\mathbf{z}, \mathbf{W}) = q_\phi(\mathbf{z})q_\theta(\mathbf{W})$, where $q_\theta(\mathbf{W})$ is Gaussian and factorises similarly across layers, including any convolutional layers preceding the capsule layers.

## 3.3 Choosing Priors: Reflecting Part-Object Assumptions

**Logistic-Normal.** Recall from Eq. (2) that each part capsule $\mathbf{c}_i$ votes for $N_{i \to j}$ objects, we can introduce randomness in their part-object connections via a Gaussian-Softmax parameterisation:

$$\mathrm{softmax}(\mathbf{z}^{(i)})_j = \frac{\exp(z_j)}{\sum_k^{N_{i \to j}} \exp(z_k)}, \qquad z_j \sim \mathcal{N}(0, 1) \qquad \text{for} \quad j = 1, 2, \ldots, N_{i \to j}, \qquad (8)$$

with all components $z_j$ sampled independently from standard Gaussian priors. The approximate posterior then takes the form: $q_\phi(\mathbf{z}^{(i)}) = \mathcal{N}(\mathbf{z}^{(i)} \mid \boldsymbol{\mu}^{(i)}, \boldsymbol{\sigma}^{(i)}) \, \forall \mathbf{c}_i \in \ell_i$. To obtain stochastic gradients of the lower bound w.r.t. the parameters $\phi$, we can parameterise samples from $q_\phi(\mathbf{z}^{(i)})$ by: $\mathbf{z}^{(i)} = f(\epsilon, \phi)$ where $f(\cdot)$ is differentiable and $\epsilon \sim \mathcal{N}(0, I)$, using the (local) reparameterisation trick [18, 20]. These priors are generally attractive since reparameterising Gaussian samples is straight forward, and they have been shown to work well in other settings such as topic models [21, 22].

**Dirichlet.** Alternatively, multi-modality over categorical events is better captured by the Dirichlet distribution [23]. We can also reduce the number of parameters as we only need to infer $\boldsymbol{\pi}^{(i)}$ rather than $\{\boldsymbol{\mu}^{(i)}, \boldsymbol{\sigma}^{(i)}\}$ for each part capsule $\mathbf{c}_i$, which is especially important in CapsNets, as explained in Section 1.1, since efficiency is a major concern. Our Dirichlet priors over $\mathbf{z}$ are defined as

$$\mathbf{z}^{(i)} = (z_1, z_2 \ldots, z_{N_{i \to j}}) \sim \mathrm{Dir}(\boldsymbol{\pi}_0^{(i)}), \qquad \boldsymbol{\pi}_0^{(i)} = (\pi_1, \pi_2, \ldots, \pi_{N_{i \to j}}), \qquad (9)$$

where $\boldsymbol{\pi}_0^{(i)}$ are the prior concentration parameters for $\mathbf{c}_i$, and the approximate posterior is then also Dirichlet distributed: $q_\phi(\mathbf{z}^{(i)}) = \mathrm{Dir}(\boldsymbol{\pi}^{(i)}) \, \forall \mathbf{c}_i \in \ell_i$. In practice, we draw Dirichlet samples via independent standard Gamma distributions over each part-object connection:

$$\boldsymbol{\gamma}^{(i)} = \{\gamma_j\}_{j=1}^{N_{i \to j}}, \qquad \gamma_j \sim \mathrm{Gamma}(\pi_j, 1), \qquad (10)$$

$$z_j = \frac{\gamma_j}{\sum_k^{N_{i \to j}} \gamma_k^{(i)}}, \qquad \text{then} \qquad \mathbf{z}^{(i)} = (z_1, z_2, \ldots, z_{N_{i \to j}}) \sim \text{Dir}(\boldsymbol{\pi}_0^{(i)}). \qquad (11)$$

This parameterisation enables significantly more efficient normalisation over objects, using a 2D transposed convolution with an identity filter to collect variable length vectors $\mathbf{z}^{(i)}$, when using convolutional voting. Unlike the Gaussian, the Gamma and Dirichlet distributions are not directly amenable to the reparameterisation trick [18, 24], so we obtain approximate pathwise gradients via the optimal mass transport (OMT) method [25]. Alternatively, we could obtain implicitly reparameterised gradients as in [26]. Both are readily available in PyTorch and Tensorflow respectively [27, 28].

### 3.4  Routing & Activating Capsules

---

**Algorithm 1** Capsule Layer with Routing Uncertainty. Returns updated object capsules $\mathbf{c}_j = \{a_j, \mathbf{M}_j\} \in \ell + 1$, given part capsules $\mathbf{c}_i = \{a_i, \mathbf{M}_i\} \in \ell$. Performs ML/MAP inference of transformation weights $\mathbf{W}$, and variational inference of latent part-object connection variables $\mathbf{z}$.

---
1: **function** CONVCAPS2D $(a_i, \mathbf{M}_i)$             ▷ input capsules from previous layer
2:      Initialise Affine Weights: $\mathbf{W}_{ij} \in \mathbb{R}^{\sqrt{d} \times \sqrt{d}} \;\; \forall i \forall j$
3:      Set Dirichlet priors: $\boldsymbol{\pi}_0^{(i)} \in \mathbb{R}^{N_{i \to j}} \;\; \forall \mathbf{c}_i \in \ell$
4:      $\mathbf{V}_{j|i} \leftarrow$ VOTE $(\mathbf{M}_i, \mathbf{W}_{ij})$    # Eq.(1)         ▷ capsules $\mathbf{c}_i$ vote for poses of capsules $\mathbf{c}_j$
5:      $\mathbf{z}_{\ell,\ell+1} \leftarrow$ SAMPLE $q_\phi(\cdot)$ $(a_i, \boldsymbol{\pi}_0^{(i)})$    # Eqs.(10-12)     ▷ sample $\mathbf{z}^{(i)}$ $\forall \mathbf{c}_i$ from approximate posterior
6:      $a_j, \mathbf{M}_j \leftarrow$ ROUTE $(\mathbf{z}_{\ell,\ell+1}, \mathbf{V}_{j|i})$    # Eqs.(12,13)     ▷ aggregate votes and activate capsules $\forall \mathbf{c}_j$
7:      **return** $\mathbf{c}_j = \{a_j, \mathbf{M}_j\}$                ▷ output capsules to next layer

---

**Global Routing.** Following from Eq. (1), part capsules $\mathbf{c}_i \in \ell$ cast votes $\mathbf{V}_{j|i}$ for object capsules $\mathbf{c}_j \in \ell + 1$, in all layers. During training we fit multivariate gaussians $\mathbf{M}_j \sim \mathcal{N}(\boldsymbol{\mu}_j, \boldsymbol{\sigma}_j)$, on each object's $d$ dimensional poses, and sample part-object connections from the approximate posterior:

$$\mathbf{z}^{(i)} \sim q_\phi(\mathbf{z}_{\ell,\ell+1}) \quad \forall \mathbf{c}_i \in \ell, \qquad \boldsymbol{\mu}_j = \frac{\sum_i \mathbf{z}_{\ell,\ell+1}^{(i)} \mathbf{V}_{j|i}}{\sum_i \mathbf{z}_{\ell,\ell+1}^{(i)}}, \qquad \boldsymbol{\sigma}_j = \frac{\sum_i \mathbf{z}_{\ell,\ell+1}^{(i)} (\mathbf{V}_{j|i} - \boldsymbol{\mu}_j)^2}{\sum_i \mathbf{z}_{\ell,\ell+1}^{(i)}}. \qquad (12)$$

The latent variables $\mathbf{z}^{(i)}$ *can* act as soft assignments depending on our choice of prior, and one could interpret the training procedure as approximating the true posterior $q_\phi^\star(\mathbf{z}|\mathcal{D}) \approx p(\mathbf{z}|\mathcal{D}, \mathbf{W})$ over all layers under the *global* minimum description length objective in Eq. (7), rather than *local* (iterative) inference of $\mathbf{z}$ in the E-step of EM routing [3] between all adjacent capsule layers. Alternatively, if for instance we let our priors on $\mathbf{z}^{(i)}$ be Beta distributed over each part-object connection, and omit the normalisation over objects, we can allow each part to route information to multiple objects at once. If one normalises over parts rather than objects, then routing closely resembles attention [29].

**Agreement & Activation.** To measure vote agreement for each object capsule, we compute the average negative entropy of its pose: $-\mathcal{H}(\mathbf{M}_j) \triangleq -d^{-1} \mathcal{H}[\mathcal{N}(\mathbf{M}_j \mid \boldsymbol{\mu}_j, \boldsymbol{\sigma}_j)]$. Averaging yields a scale invariant measure w.r.t. the number of pose parameters $d$. Agreement is weighted by the support for each object capsule, which is the amount of data received from its parts: $-\mathcal{H}(\mathbf{M}_j) \sum_i \mathbf{z}_{\ell,\ell+1}^{(i)}$. Next, consider a Binomially distributed random variable $S_j \sim \mathcal{B}(N_i, N_j^{-1})$, describing the assignment of $N_i$ parts to $N_j$ objects with probability $N_j^{-1}$. The expected amount of data each object receives in a given layer is then $\mathbb{E}(S_j)$. We can use this value to normalise and offset the entropy term, which automatically scales logits according to the number of capsules in each layer:

$$a_j \triangleq \frac{-\boldsymbol{\eta}_j \mathcal{H}(\mathbf{M}_j) - \mathbb{E}(S_j)}{\mathbb{E}(S_j)} = -\frac{\boldsymbol{\eta}_j}{\mathbb{E}(S_j)} \mathcal{H}(\mathbf{M}_j) - 1, \qquad \boldsymbol{\eta}_j \triangleq \sum_i \mathbf{z}_{\ell,\ell+1}^{(i)}, \qquad (13)$$

$a_j$ is then activated using the logistic function. In simple terms, if the uncertainty among votes is **high** — i.e. low negative entropy and *poor* agreement — assigning more data to capsule $j$ *decreases* its activation. Alternatively, if the uncertainty among votes is **low** — i.e. high negative entropy and *good* agreement — assigning more data to capsule $j$ *increases* its activation significantly. Activating capsules in this way simply encourages the model to meet the *agreement* and *support* activation criteria implicitly, but does not enforce them explicitly via learned $\beta$ thresholds as in EM routing [3].

Table 1: Comparing viewpoint-invariance on SmallNORB. Performances are matched on familiar viewpoints, before testing on novel. Results from 3 random seeds on architectures $\{f_0, t_1, t_2, t_3, t_4\}$.

| Method (Viewpoints) | **Azimuth** (Acc. %) $\mathcal{A}_{\text{train}}$ | $\mathcal{A}_{\text{test}}$ | **Elevation** (Acc. %) $\mathcal{E}_{\text{train}}$ | $\mathcal{E}_{\text{test}}$ | # Param |
|---|---|---|---|---|---|
| Baseline CNN [3] | 96.3 | 80.0 | 95.7 | 82.2 | 4.2M |
| CNN (AvgPool) [12] | 91.5 | 78.2 | 94.3 | 82.28 | 0.15M |
| Our EM-Routing | 96.29±0.02 | 87.1±0.42 | 95.71±0.02 | 87.9±0.39 | 0.17M |
| SR-Caps [12] | 92.38 | 80.14 | 94.04 | 84.09 | 0.75M |
| STAR-Caps [11] | 96.3 | 86.3 | - | - | 0.32M |
| EM-Routing [3] | 96.3 | 86.5 | 95.7 | 87.7 | 0.31M |
| VB-Routing [13] | 96.29 | 88.6 | 95.68 | 88.4 | 0.17M |
| $\{32, 8, 8, 8, 5\}$ | 96.3±0.03 | 89.12±0.7 | 95.68±0.04 | 89.64±0.49 | 0.06M |
| $\{64, 8, 16, 16, 5\}$ | 96.3±0.02 | 91.06±0.31 | 95.7±0.02 | 91.01±0.26 | 0.14M |
| $\{64, 16, 16, 16, 5\}$ | 96.29±0.03 | 91.41±0.46 | 95.7±0.03 | 91.36±0.4 | 0.22M |
| $\{128, 16, 32, 32, 5\}$ | 96.3±0.02 | 91.85±0.42 | 95.71±0.03 | 92.03±0.21 | 0.58M |

**Capsule $L_2$ Norm.** Alternatively, we can activate capsules by computing the Frobenius norm of the mean votes for object poses $||\boldsymbol{\mu}_j||_F$, then squashing it to a sensible $(0, 1)$ range [2]. This encodes agreement in the norm of the poses and offers a considerable speedup at a performance cost.

## 4 Experiments

In this study, we focus on demonstrating that our method enhances capsule properties and outperforms previous approaches on challenging pose-aware tasks used in CapsNet literature (Sections 4.1, 4.2 and 4.4), whilst being more computationally efficient (see Figure 4 for runtime comparisons).[1]

**Network Architecture.** To ensure fair and direct comparisons with previous work, we use identical CapsNets to EM routing [3]. A single $5 \times 5$ `Conv` layer with $f_0$ filters and stride 2 precedes four capsule layers. The `PrimaryCaps` layer transforms $f_0$ feature maps into $t_1$ capsule types, each having $H \times W$ number of capsules with $4 \times 4$ poses. Next, two $3 \times 3$ `ConvCaps` layers with $t_2$ and $t_3$ output capsule types, using strides 2 and 1. The last `ConvCaps` layer outputs $t_4$ class capsules, and shares weights across spatial dimensions [3]. Let $\{f_o, t_1, t_2, t_3, t_4\}$ denote the complete architecture. In all experiments, we use Adam [30] with default parameters and a batch size of 128 for training.

**Priors.** To show our method works well in the general case, we set the priors to be as uninformative as possible for all the benchmark results presented, i.e. flat Dirichlet: $p(\mathbf{z}^{(i)}) \sim \text{Dir}(\mathbf{1}_{N_{i \to j}}) \, \forall \mathbf{c}_i \in \ell, \, \forall \ell$. These priors explicitly assume that each part capsule $\mathbf{c}_i$ is equally likely to belong to any object it votes for, with any level of certainty. Nonetheless, we conducted experiments to test sensitivity to the choice of prior, as presented in Figure 4. We observe tighter bounds for priors with central peaks, meaning that sampled part-object connections are closer to uniform over objects. Although tighter bounds are not always better [31], this suggests that parts prefer to spread their vote amongst multiple objects in CapsNets, which is reminiscent of Dropout's effect on NN weights [32].

**Inference.** In all benchmark results, we perform a deterministic inference at test time without sampling $\mathbf{z}$, by using the posterior means $\mathbf{z}^\star = \mathbb{E}[q_\phi^\star(\mathbf{z}_{\ell, \ell+1})] \, \forall \ell$, to compute predictions $\mathbf{y}^\star = \arg\max_{\mathbf{y}} p(\mathbf{y}|\mathbf{x}, \mathbf{z}^\star, \mathbf{W})$. Alternatively, we can draw $T$ Monte Carlo samples of part-object connections from the approximate posterior, and calculate the predictive entropy:

$$\mathcal{H}(\widehat{\mathbf{y}}|\mathbf{x}, \mathbf{z}, \mathbf{W}) = -\sum_{k=1}^{K} \widehat{\mathbf{y}}_k \log \widehat{\mathbf{y}}_k, \qquad \widehat{\mathbf{y}} \approx \frac{1}{T}\sum_{t=1}^{T} p(\mathbf{y}|\mathbf{x}, \mathbf{z}^t, \mathbf{W}), \qquad \mathbf{z}^t \sim q_\phi^\star(\mathbf{z}|\mathcal{D}). \quad (14)$$

Under full posterior learning: $q_{\phi,\theta}(\mathbf{z}, \mathbf{W})$, the pose transformation matrices $\mathbf{W}$ are also sampled. Although the model is partially Bayesian, we observe predictive entropies on out-of-distribution dataset samples (AffNIST, FashionMNIST) to be consistent with model uncertainty representation as shown in Figure 3. We also observe entropic predictions on more challenging SmallNORB viewpoints as we vary azimuth, whilst holding the lowest/highest elevation viewpoints fixed (see Figure 2).

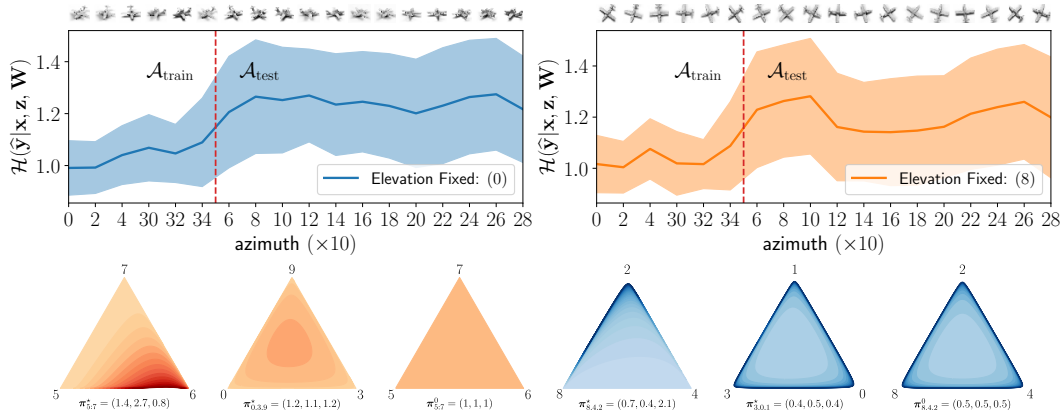

Figure 2: (Top row) Predictive entropies when varying azimuth viewpoints whilst holding lowest/highest elevations fixed on SmallNORB. Obtained with 10 MC samples using $\{32, 8, 8, 8, 5\}$. (Bottom row) Example posterior parameters $\pi^\star$ from two random penultimate layer capsules of networks trained under different Dirichlet priors $\pi_0$, simplex corners represent SVHN digit classes.

## 4.1 Generalisation to Novel Viewpoints

**Viewpoint-Invariance.** SmallNORB [33] consists of grey-level stereo $96 \times 96$ images of 5 objects: each given at 18 different azimuths (0-340), 9 elevations and 6 lighting conditions, with 24,300 training and test set examples. As in [3], we standardise the images and resize them to $48 \times 48$. During training we take $32 \times 32$ random crops, and centre crops at test time. We train on training set images with azimuths: $\mathcal{A}_{\text{train}} = \{300, 320, 340, 0, 20, 40\}$, denoted as *familiar* viewpoints, and test on test set images containing *novel* azimuths: $\mathcal{A}_{\text{test}} = \{60, 80, \ldots, 280\}$. Similarly, for the elevation viewpoints we train on $\mathcal{E}_{\text{train}} = \{30, 35, 40\}$, and test on $\mathcal{E}_{\text{test}} = \{45, 50, \ldots, 70\}$. As reported in Table 1, we observed notable performance improvements in viewpoint-invariance over previous CapsNets, and significant improvements over CNNs. Additional results on the standard SmallNORB train/test splits are found in Table 2.

Table 2: SmallNORB test error (%), results from 3 random seeds.

| Method | SmallNORB | |
|---|---|---|
| | Error (%) | # Param |
| Baseline CNN [3] | 5.2 | 4.2M |
| Our CNN | 5.6±0.12 | 2.4M |
| Our ResNet-20 | 2.7±0.11 | 0.27M |
| Our EM-Routing | 1.9±0.15 | 0.17M |
| Dynamic [2] | 2.7 | 8.2M |
| FRMS [10] | 2.6 | 1.2M |
| FREM [10] | 2.2 | 1.2M |
| STAR-Caps [11] | 1.8 | 0.25M |
| EM-Routing [3] | 1.8 | 0.31M |
| VB-Routing [13] | 1.6 | 0.17M |
| $\{32, 8, 8, 8, 5\}$ | 2.2±0.08 | 0.06M |
| $\{64, 16, 16, 16, 5\}$ | 1.5±0.10 | 0.22M |
| $\{64, 8, 16, 16, 5\}$ | 1.4±0.09 | 0.14M |

## 4.2 Affine Transformation Robustness

Table 3: MNIST to AffNIST generalisation error (%). (†) unsupervised learning.

| Method | MNIST | AffNIST |
|---|---|---|
| | Test Error (%) | |
| Baseline CNN [3] | 0.8 | 14.1 |
| BCN [34] | 2.5 | 8.4 |
| Dynamic [2] | 0.77 | 21 |
| G-Caps [35] | 1.58 | 10.1 |
| Sparse-Caps[36] | 1.0† | 9.9 |
| SCAE [4]† | 1.5 | 7.79 |
| EM-Routing [3] | 0.8 | 6.9 |
| Aff-Caps [37] | 0.77 | 6.79 |
| $\{32, 8, 8, 8, 10\}$ | 0.8±0.01 | 5.02±0.28 |
| $\{64, 8, 16, 16, 10\}$ | 0.79±0.01 | 4.17±0.3 |
| $\{64, 16, 16, 16, 10\}$ | 0.78±0.02 | 3.88±0.34 |
| $\{128, 16, 32, 32, 10\}$ | 0.8±0.02 | 3.46±0.19 |
| $\{128, 16, 32, 32, 10\}$ | 0.28±0.01 | 2.31±0.03 |

**Out-of-Distribution Generalisation.** In this study we demonstrate our model's robustness to affine transformations using the AffNIST dataset. AffNIST consists of MNIST images which have been uniquely transformed by 32 random affine transformations per image. Training is performed on the MNIST training set, and we test generalisation performance on the AffNIST test set containing 320,000 examples. AffNIST images are $40 \times 40$ so for training we pad MNIST images, randomly placing the digits on $40 \times 40$ black backgrounds as in works we compare to [2, 13]. Our models were never trained on AffNIST, and no further data augmentation was used. As shown in Table 3, we observed performance improvements over previous CapsNets, and significantly so over CNNs. Increasing the number of capsules used in our method also leads to better generalisation performance.

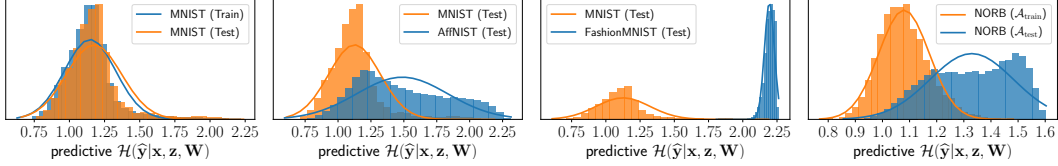

Figure 3: Histograms of predictive entropies on in- and out-of-distribution test examples. Results obtained with 10 MC samples from $q^\star_\phi(\mathbf{z}|\mathcal{D})$ using our $\{64, 8, 16, 16, 10\}$ model.

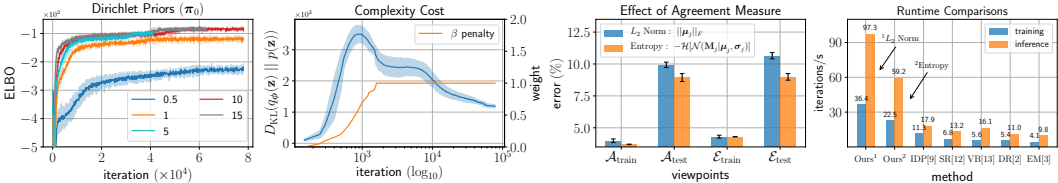

Figure 4: Effect of symmetric Dirichlet priors on the tightness of the ELBO over 3 runs on SVHN 10K, and complexity cost (KL) with $\beta$ weight penalty throughout training (Left). Comparing capsule activation methods on SmallNORB viewpoint performance, and runtimes (CIFAR-10) of 5 open source routing methods ran on 2 Titan Xp GPUs, using the same $\{128, 16, 16, 16, 10\}$ model (Right).

## 4.3 Limited Training Data Regime

**Sample Efficiency.** Rather than artificially applying perturbations, we leverage the natural range of geometric variation in the SVHN dataset [39] to verify robustness and generalisation performance on real data. We follow the experimental setup of Equivariant Transformers [38], and train models with random 10K and 20K subsets of the original training set of 73,257 examples, and evaluate on the test set (26,032). As shown in Table 4, our CapsNets are quite sample efficient in the limited training data regime, offering modest improvements over STN/ETN baselines in [38], and significantly so over CNNs. Sample efficiency is critical in real world tasks where data is limited. Interestingly, we observe smaller improvements over baselines

Table 4: Comparing SVHN test error (%) with limited training data, from 3 random seed runs.

| Method | SVHN | | # Param |
|---|---|---|---|
| (#Train) | 10K | 20K | |
| ResNet-18 [38] | 9.83 | 7.90 | 2.7M |
| ResNet-34 [38] | 8.73 | 7.05 | 5.2M |
| Our CNN | 9.4±0.25 | 7.7±0.21 | 2.4M |
| ResNet-18 (STN) | 9.10 | 7.17 | 2.8M |
| (ETN) | 7.81 | 6.37 | |
| ResNet-34 (STN) | 8.60 | 6.91 | 5.3M |
| (ETN) | 7.72 | 5.98 | |
| $\{32, 8, 8, 10\}$ | 7.7±0.05 | 6.5±0.04 | 0.06M |
| $\{64, 16, 16, 16, 10\}$ | 7.5±0.21 | 5.9±0.26 | 0.22M |
| $\{64, 8, 16, 16, 10\}$ | 7.0±0.15 | 5.9±0.11 | 0.15M |

as more training data is used, suggesting that model choice is less important given enough data.

## 4.4 Performance Under Feature Occlusion

Table 5: Comparing MultiMNIST test error and exact match ratio (MR) error. (†) dagger denotes results from using DiverseMultiMNIST.

| Method | MultiMNIST | | # Param |
|---|---|---|---|
| (Test) | Error (%) | MR (%) | |
| Baseline CNN [2] | 8.01 | - | 24.6M |
| Baseline CNN [9]† | - | 15.2 | 19.6M |
| Dynamic [2] | 5.2 | - | 8.2M |
| IDP-Attention [9]† | - | 8.83 | 42M |
| Aff-Caps [37] | 4.51 | - | 8.2M |
| $\{64, 8, 16, 16, 10\}$ | 3.3±0.07 | 7.2±0.21 | 0.15M |
| $\{128, 16, 16, 16, 10\}$ | 2.4±0.11 | 4.7±0.18 | 0.23M |
| $\{128, 16, 32, 32, 10\}$ | 1.8±0.09 | 3.4±0.17 | 0.58M |

**Overlapping Digits.** In this study we empirically demonstrate that our method is resilient under feature occlusions (which is a source of uncertainty). To that end, we replicated the experiment setup in [2], and trained our shallow models on the MultiMNIST dataset by generating occluded digit pairs on the fly. Digit pairs are formed by shifting each MNIST digit by up to 4 pixels in each direction, then adding them together. No further data augmentation was used. Our models were trained/validated on 60M overlapping digit pairs, and tested on 10M. Table 5 reports both lower test error and exact match ratio (MR) error compared to previous work. See Figure 5 for illustrations.

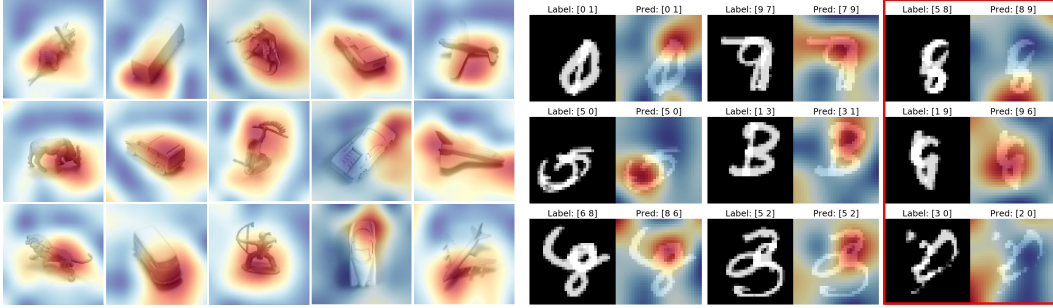

Figure 5: Explanatory heat maps of predictions by our models trained on SmallNORB (Left) and MultiMNIST (Right). Obtained by upsampling the posterior means of the part-object connections $\mathbf{z}^{\star}$ in the class capsule layer, up to the input size: yielding attention-like explanations of predictions.

## 5 Related Work & Conclusion

**Variational Inference.** Our work lies at the intersection of CapsNets and variational Bayesian learning. Variational Inference (VI) has its roots in statistical physics [40, 41], leading to seminal work in the early-1990s [15] which offered a MDL [14] perspective on VI in NNs. VI was later formalised more generally in a series of important works [42, 43, 44, 16]. More recently, practical strategies for calculating biased/unbiased Monte Carlo gradients of variational objectives in deep NNs have been proposed [17, 19], which are complemented by ideas from deep generative modelling such as the reparameterisation trick [18, 24]. NNs with Dropout [32] have also been interpreted as being approximately Bayesian [45, 46], and are widely used to estimate uncertainty [47, 48, 49, 50].

**Capsule Networks.** Initial work on capsules began with the transforming autoencoder [1]. Other successful variants have since then been proposed, notably: Dynamic routing [2], EM routing [3], and stacked capsule autoencoders (SCAE) [4], all of which achieved state-of-the-art performance in pose-aware tasks. Much follow-up work focuses on algorithmic variants of *local* routing or in scaling up CapsNets: VB routing [13], KDE [10], Spectral [51], Subspace-Caps [52, 53]. Other interesting works improve on the equivariance properties of CapsNets directly using Group theory [35, 54], which is on the contrary to our approach, as we do not impose any specific equivariance restrictions into the model. Geometric approaches have also been explored by [55, 56, 57], extending CapsNets to work with point clouds and in 3D. Related work on probabilistic interpretations of CapsNets is limited, with the notable exception of [58] which considers a fully generative perspective of SCAE [4] that is unsupervised, in contrast to the discriminative probabilistic model with capsule structure presented in this paper. Our work builds primarily on both *local* EM/VB routing [3, 13]–to which we provide a *global* alternative view using VI tools–and other recent non-iterative routing methods: Attention routing [59], STAR-Caps [11], Self-Routing [12], and Inverted Dot-Product routing [9]. Modifications in some of the latter methods have led to ambiguities regarding what qualifies as a CapsNet, as opposed to CNNs with attention. As explained in Section 1.1, this occurs whenever the fundamental premises of CapsNets are implicitly or explicitly altered, and their properties are not carefully verified or retained. With that in mind, we demonstrate empirically that our proposed end-to-end probabilistic approach leads to performance enhancements in benchmark pose-aware tasks commonly used in CapsNet literature, whilst being more computationally efficient.

### 5.1 Conclusion

In this paper we propose to replace inefficient *local* iterative routing with variational inference of a posterior on part-object connections in a probabilistic capsule network, leading to a significant speedup (Figure 4). In this way, we encourage *global* context to be taken into account when routing information, by introducing *global* latent variables which have direct influence on the objective function, and are updated discriminatively in accordance with the minimum description length principle. To facilitate comparisons, we developed our method in a discriminative setting, and performed a thorough evaluation on pose-aware tasks, demonstrating enhanced capsule properties over previous iterative and non-iterative routing methods. We believe further exploration of CapsNets as deep latent variable models (DLVMs) [24, 60, 61], to be a promising future research direction.

## Broader Impact

With the advent of Deep Learning, the computational requirements in the field have increased significantly due to the ever increasing scale of our models. The environmental impact of training or deploying such models is therefore at an all time high. This raises concerns regarding the sustainability of our current practices, as the technologies we help develop are slowly integrated into all areas of society. Although it is important to continue on this path of discovery, we feel that an important shift towards efficiency is sorely needed. Concretely, the development of smaller scale models which are more robust and sample efficient, could significantly reduce the environmental impact of our technology with small sacrifices in performance. In general, we believe this can be achieved by introducing richer inductive priors into our models, which in turn require fewer examples to learn from, i.e. leading to increased sample efficiency. With that in mind, Capsule Networks have previously shown to possess superior generalisation properties than conventional CNNs in certain tasks, and in our work we enhance these properties further whilst being more computationally efficient than previous iterative routing methods. We also demonstrated competitive performances on sample efficiency tasks, which have broad applicability to limited data domains such as medical. When these properties are enhanced even further, they have the potential to make a significant positive impact on our societies by increasing the sustainability and efficiency of our machine learning models.

## Acknowledgments and Disclosure of Funding

We would like to gratefully acknowledge the support of NVIDIA Corporation with the donation of GPUs used for this research. We also thank Francesco Caliva and Lewis Smith for fruitful discussions.

## Footnotes

[1] Code available at: `https://github.com/fabio-deep/Routing-Uncertainty-CapsNet`

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
