[Reviews · NeurIPS 2020]

Review 1

Summary and Contributions: Capsule networks are capable of representing and learning the viewpoint variations of objects; however, the common iterative routing mechanisms [2, 3] used between the adjacent capsule layers are inefficient. This paper proposes a global view based on representing the uncertainty in part-object (part-whole) relationships by approximating a posterior distribution over part-object connections. The routing uncertainty (uncertainty in assembling object parts between capsule layers) might arise from sources such as feature occlusions due to observed viewpoints, noisy data, and ambiguous poses of some objects such as spherical object parts. In the proposed approach, the local routing iterations are replaced with variational inference of capsules connections, which leads to a significant speedup. The global context during routing is encouraged through using global latent variables which directly influence the objective function, and get updated during training in accordance with the minimum description length (MDL) [14]. Experiments show that the proposed approach provides improvements over previous related approaches.

Strengths: - The paper is clear and well-written. It provides informative explanations including different figures and visualizations to illustrate the problem, the motivation, and the proposed framework. - The paper tackles a major drawback of capsule networks, which is the inefficient iterative routing used between the adjacent capsule layers. To speed up the routing, the proposed framework replaces the local routing iterations with variational inference of capsules connections. In addition, a global context is encouraged during routing through utilizing global latent variables which directly affect the objective function. - The proposed framework was evaluated through a different set of experiments including, comparisons of viewpoint-invariance and testing accuracies on SmallNORB dataset with other related work, robustness to affine transformations on MNIST/AffNIST datasets, and the testing performance on SVHN dataset.

Weaknesses: Although the paper is well-motivated and provided a good introduction to the drawbacks of capsule networks, it did not include comprehensive empirical results to show the efficacy of the proposed approach. Specifically, the computational complexity of the routing mechanisms was mentioned as one of the bottlenecks of capsule networks (lines 32-35) in addition to emphasizing that the proposed routing approach provides significant speedup (line 60), however, the paper only provided a figure (fig. 4, Top right) that shows runtime comparisons on SmallNORB dataset to two approaches EM[3] and VB[8]. The computational complexity of the capsule routing poses a major limitation in both the training and the testing stages, in addition, some studies (e.g. [10]) show that it is challenging to train a baseline capsule network such as EM[3] on a relatively more complex image datasets such as CIFAR100 or datasets with high resolution and natural scenes such as ImageNet. Therefore, solving or mitigating this limitation plays a crucial factor in advancing the applicability of capsule networks on datasets with higher resolution images (e.g. ImageNet) or on advanced computer vision problems such as object detection. In addition, some related work such as EM[3] and STAR-Caps[10] presented some results on more complex image datasets such as CIFAR10 in EM[3] and STAR-Caps[10], and on CIFAR100 and ImageNet datasets as in STAR-Caps[10].

Correctness: The method and the empirical methodology appear to be correct

Clarity: The paper is clear, and well-written.

Relation to Prior Work: Relation to prior work was discussed in the paper.

Reproducibility: Yes

Additional Feedback: - Can you provide more detailed runtime comparisons to other related approaches (e.g. EM[3] and STAR-Caps[10]), in addition to testing performance on other datasets such as CIFAR10/100 (explained in the Weaknesses section above) - In figure 4, the second plot (top left), seems to be missing some information, please comment and provide updates.


Review 2

Summary and Contributions: This paper proposes a learning based method as a replacement of local routing algorithm in capsule networks. The new method is inspired by globally learning part-object connection through variational inference. The performance of the proposed method is verified on unseen viewpoints of SmallNorb dataset, MNIST and AffMNIST. Also the performance of the model under limited training samples is evaluated for SVHN dataset.

Strengths: -The idea of improving or removing the routing procedure from capsule networks is not new and has been studied in the past 2 years as clearly covered by authors in the related work section but their perspective to the problem is interesting. - Based on the experiments, the proposed method leads to better or on par performance with the state of the art method but with fewer parameters.

Weaknesses: 1- in the line 57, authors correctly mention that one of the sources of uncertainty in assembling objects via a composition of parts is "feature occlusions due to observed viewpoints" . But unfortunately there is no experiment supporting that the proposed method is more efficient under feature occlusion. IT would be nice if you could add experiment on MultiMNIST to showcase the effectiveness of the proposed method on occluded images. 2- Based on the line 121 in the section 3.2, the approximate posterior of part-object connections between capsule of consecutive layers of L and L+1 are learned by a network q_{delata}(z_{l,l+1} but it is not clear what the architecture of these networks are. So how is the number of parameters in this method is less than the baseline method considering both have the same number of convolution and capsule layers and the proposed method has an extra network between each 2 layer to infer Z_{L, L+1} 3 - One of the motivation for this method is memory bottleneck of original capsule network, but there is no experiment to support the effectiveness and the practicality of the method on more complex tasks. At least beyond 10 classes.

Correctness: yes

Clarity: It would be nice if the authors could summarize the proposed method in an algorithm. It will simplify the comparison with other methods. Please clarify about the point 2 of weakness

Relation to Prior Work: Yes

Reproducibility: No

Additional Feedback: The author(s) responded all of my questions.


Review 3

Summary and Contributions: This paper proposes a new alternative global routing in capsule networks, based on representing the inherent uncertainty in part-object composition. The models are evaluated on viewpoint-invariance and affine transformation robustness tasks, showing improvements over previous approaches.

Strengths: This paper provides a sufficient theoretical understanding of variational inference (VI), and generally summarizes the key assumptions of capsule networks.

Weaknesses: - Motivation and insight is unclear. 1) As the title of this paper says, the main contribution may be introducing routing uncertainty in capsule networks with VI. But VB-routing [1] also studied uncertainty in their method with VI. Please explain the connection and difference with VB-routing. 2) Section 4 ‘Related Works’ is insufficient. There are many works which focus on the speeding up the routing procedure in capsule networks, such as [2]. Please discuss more detailly about the different motivations/understandings/perspectives of these methods, and provide the main insights of your method for this field. - The paper is hard to understand. 1) There many equations in this paper. I think an Algorithm cell will help readers understand the overall algorithm, as in [1] and [3]. 2) The tables and figures are poorly interpreted. For example, in Table 1, what’s the difference between ‘EM-Routing’, ‘Our EM-Routing’ and ‘{32, 8, 8, 8, 5}’. Groups of figures in Figure 3 and Figure 4 also need to be explained separately. 3) In Eq. (12), the subscripts need to be better organized. \sum_{i \in l} z_{l, l+1}, I think this summation is over multiple ‘i’s in the ‘l’, but there is no ‘i’ subscript in ‘z’ term. This summation may be confusing. 4) In L169, does the ‘N(M | \mu, \sigma)’ mean the posterior likelihood of M, if so, it is a scalar, how do you compute the entropy. - Ablation analysis. There are many components, such as different VIs, different priors. I think an ablation analysis would be better. [1] Capsule Routing via Variational Bayes. AAAI 2020. [2] Improving the Robustness of Capsule Networks to Image Affine Transformations. CVPR 2020. [3] Matrix Capsules with EM Routing. ICLR 2018.

Correctness: Yes

Clarity: Please refer to ‘Weaknesses’

Relation to Prior Work: Please refer to ‘Weaknesses’

Reproducibility: Yes

Additional Feedback: Please address my detailed comments mentioned in ‘Weaknesses’.


Review 4

Summary and Contributions: Authors introduce a variational inference framework for optimizing and inferring capsule networks and their routing parameters. This work is based on the em routing matrix capsules framework. They essentially remove the em routing and suggest a global optimization for minimum description length. There are several interesting and novel contributions in this work. Their method no longer has iterative local routing steps but rather can optimize for connections (z) globally via variational inference. They approximate the p(z|D) (connection distribution) via a parameterized neural network between each two capsule layers. As for prior they argue that Dirichlet distribution is better suited to this task in compare to normal distribution both due to multi modality and fewer number of parameters. Then they infer the parameters of the Dirichlet using the aformentioned NN between capsule layers. They define the activation of capsules by weighing the agreement (average negative entropy) by the support that it receives (some of z). Then they use a binomial distribution to offset and normalize the activations in one one layer. This enables them to have an activation measure that does not rely on specific hyper parameters (beta in em capsule routing agreements). They introduce an end-to-end free energy optimizer which minimizes the description length. However in practice they optimize for connections z via variational inference and use MAP for the weights W. They provide empirical results which shows their superiority in terms of performance and viewpoint generalization.

Strengths: Their framework provides a more coherent and theoretically sound method than the original em capsules. e.g. they directly optimize the mdl loss. They can optimize routing parameters globally. They achieve exciting and sota results on norb, mnist and svhn. They enhance the viewpoint generalizability of the baseline capsules to several degrees. Recap: They introduce a novel and significant reformulation of CapsuleNetwork which opens the path for further research. Their global and non-iterative agreement finding improves the speed and memory cost of this networks. In this VI framework, several new paths can be explored by community. Moreover, it improves the classification results of capsule networks while improving their viewpoint generalization.

Weaknesses: The writeup should be improved significantly. The paper is quite hard to follow specially with an eye for details. I am not sure if I got all the details even after several passes. Some examples are: fig. 1 is too small and crowded. Some of the symbols used in it are not explained anywhere. It is trying to convey too much information in too small a space. Related work section is too brief, covering only capsule related work and missing VI related work. The setup for network q is not explained, how many layers, structure, non linearity, etc. From page 4 and fig.1 there seems to be a network q, but in the experiments section there is no mention of it and it seems to ignore that and use a flat dirichlet always. The independence assumption in 5, although convenient is not intuitively correct. The kl equations is between P(z|D) and q(z), the optimization is on p(D|z), there is no equation between p(z|d) and p(d|z). In general the equations are scattered around and their connections are not straight forward. nit: the dimension is nicer to be sqrt(d)xsqrt(d) rather than d/2xd/2 for W and M_i. It results in same number of parameters as a vector of d.

Correctness: Most probably.

Clarity: No, it should be improved.

Relation to Prior Work: Medium. The related work section is brief and not informative. Also understanding this work relies on reading and understanding EM Matrix Capsules paper. The most relevant sections of em routing (like the em algorithm or their minimum description length approximation) is better to be included in paper to clarify the comparisons. For example unless someone is quite familiar with the em paper they won't get the reference about beta in the soft assignment for measuring activation and agreement.

Reproducibility: Yes

Additional Feedback: Please improve the writing. This is a potentially significant work with many opportunities for further research. Unless the community can understand and reproduce the results it will not be useful. Thanks.

[Author Response · NeurIPS 2020]

We would like to thank the reviewers for their comments and positive outlook on the paper. We are encouraged that all reviewers find our proposed method, claims and empirical methodology to be correct (**R1**, **R2**, **R3**, **R4**). **R1** found our paper to be well-motivated in tackling a major efficiency drawback of capsule networks, informative and clear, whereas **R2** found it to provide an interesting new perspective. **R3** acknowledged our theoretical understanding of variational inference and capsule networks. **R4** stated that we introduce a novel and significant reformulation of capsule networks which is more coherent and theoretically sound than previous work, achieving exciting state of the art results and enhancing the viewpoint generalizability of capsules by several degrees. We hope our response clarifies all concerns.

**[R1]** **1. More runtime comparisons.** As requested, we conducted more extensive runtime comparisons with the 5 most prominent and publicly reproducible related works. For fairness, we use the same $\{128, 16, 16, 16, 10\}$ architecture and replace the routing mechanism. We use Pytorch, 2 Titan Xp GPUs and a batch size of 64. As depicted on the right, our method offers considerable speedups over previous works, whilst enhancing performance on pose-aware tasks (see paper).

**[R2]** **2. Feature occlusion experiments (MultiMNIST).** We thank the reviewer for the valueable suggestion. We empirically demonstrate that, unlike previous methods, modelling uncertainty over part-object connections yields significantly more resilient capsnets under feature occlusion (which is a source of uncertainty). We replicated the experiment setup in [2], and trained our shallow $\{128, 16, 16, 16, 10\}$ model on MultiMNIST by generating occluded digit pairs on the fly. We trained for 300 epochs on $\simeq 18$M training examples. Table 1 reports both test accuracy and exact match ratio (MR). As shown, our method outperforms previous work by a large margin using fewer parameters.

**[R1] [R2]** **3. Evaluation on CIFAR-10/100.** Although our work is focused on enhancing capsule network properties in pose-aware tasks, we evaluated our method on CIFAR-10/100 as suggested. We borrow the setup and baselines from [9] and compare with the most prominent previous works which are publicly reproducible (see Table 1). For fair comparisons, we used the shallow model $\{128, 16, 32, 32, 10\}$ described in Section 5, and baseline CNNs of equal depth. By replacing the single `Conv` layer stem with a ResNet-20 backbone we achieve $93.1\%$ ($1.92$M) on CIFAR-10, and $72.4\%$ ($2.01$M) on CIFAR-100. With a *thinner* $\{32, 8, 8, 8, 10\}$ model we can achieve $90.5\%$ on CIFAR-10 using only $0.1$M parameters.

**[R2] [R4]** **4. Further details on inference networks** $q_\phi(\cdot)$**.** This will be rectified in the final version. For clarity, each $q_\phi(\cdot)$ is simply a single layer perceptron with sofplus non-linearities that takes the activations of part capsules $\mathbf{a}_i$ and outputs the parameters $\boldsymbol{\pi}^{(i)}$ of the approximate Dirichlet posterior on the part-object connections. For **R2**, the number of parameters is kept small both thanks to our choice of Dirichlet prior as discussed in Section 3.3, and our use of fewer capsules than previous work whilst achieving better performance, i.e. at most $\{128, 16, 32, 32, 10\}$.

Table 1: CIFAR10/100 & MultiMNIST.

| Method | Test Acc. (# params) | |
|---|---|---|
| | CIFAR-10 | CIFAR-100 |
| Baseline CNN | 82.2 (2.4M) | 51.4 (2.4M) |
| Baseline CNN [9] | 87.1 (18.9M) | 62.3 (19M) |
| Dynamic [2] | 84.1 (7.9M) | 56.9 (32M) |
| EM-Routing [3] | 82.2 (0.5M) | 37.7 (0.5M) |
| IDP-Attention [9] | 85.1 (0.6M) | 57.3 (1.5M) |
| VB-Routing [12] | 86.2 (0.4M) | 58.4 (0.5M) |
| Ours | 88.3 (0.57M) | 63.4 (0.65M) |

| Method | MultiMNIST (#params) | |
|---|---|---|
| | Test Acc. (%) | Test MR (%) |
| Baselines [2][9] | 91.9 (24.6M) | 84.8 (19.6M) |
| Dynamic [2] | 94.8 (8.2M) | - |
| IDP-Attention [9] | - | 91.17 (42M) |
| Aff-Caps [42] | 95.49 (8.2M) | - |
| Ours | **97.96** (0.23M) | **96.4** (0.23M) |

**[R3]** **5. Explain connection & difference to VB-Routing.** Our method is related to VB-Routing but fundamentally different. In VB-Routing the authors perform closed-form variational-EM updates, which are still iterative and *local*, just like EM-Routing. Therefore, VB-Routing still suffers from the efficiency drawbacks mentioned in Section 1.1. In our case, we perform *global* variational inference of part-object connections in a fully probabilistic capsule network, that is locally non-iterative and is trained end-to-end under a single globally coherent minimum description length objective (Eq. 7). Lastly, the VB-Routing framework does not provide predictive uncertainty estimates, whereas our work is the first to do so in the capsule domain to the best of our knowledge.

**[R3]** **6. Provide main insights of the method for the field.** As aptly summarised by **R4**, we provide a more coherent and theoretically sound capsule routing framework by directly optimising an end-to-end MDL objective. Our approach offers a significant speedup over previous methods, provides uncertainty estimates, and is the first non-iterative non-local routing method to enhance capsule network properties such as viewpoint generalisation by several degrees.

**[R4] [R3]** **7. Improvements to paper clarity & related work.** We thank the reviewers for the constructive feedback, and we agree that the exposition can be difficult to follow. We will make Figures 1 & 4 (**R1**) more legible, and rearrange the equations. Given the availability of an extra page in the camera-ready version, we will include an algorithm cell and an additional paragraph on related work, incorporating the reference to (Gu, J. and Tresp, V., 2020) mentioned by **R3** and prior work on variational inference. Key details from EM-Routing will be added to aid in general understanding (**R4**). Lastly, the tables will be made clearer, better indicating the differences between methods. For **R3**, 'Our EM-Routing' simply denotes our implementation of EM-Routing [3], and '$\{32, 8, 8, 8, 5\}$' denotes a variant of our architecture.

[Meta-Review · NeurIPS 2020]

Four expert referees are overall positive about the paper. Main pros are theoretical elegance and good performance, main concerns include relatively poor presentation and incomplete discussion of related work. The authors addressed the concerns adequately in the rebuttal, thus at this point I recommend acceptance. The authors are strongly encouraged to address the reviewers' concerns in the final version of the paper.